# Tumor-Infiltrating Immune Cells and HLA Expression as Potential Biomarkers Predicting Response to PD-1 Inhibitor Therapy in Stage IV Melanoma Patients

**DOI:** 10.3390/biom14121609

**Published:** 2024-12-16

**Authors:** Barbara Hegyi, Kristóf György Csikó, Tímea Balatoni, Georgina Fröhlich, Katalin Bőcs, Erika Tóth, Anita Mohos, Anna Rebeka Neumark, Csenge Dorottya Menyhárt, Soldano Ferrone, Andrea Ladányi

**Affiliations:** 1Department of Chest and Abdominal Tumors and Clinical Pharmacology, National Institute of Oncology, H-1122 Budapest, Hungary; hegyi.barbara@oncol.hu (B.H.); csiko.kristofgyorgy@gmail.com (K.G.C.); 2National Tumor Biology Laboratory, National Institute of Oncology, H-1122 Budapest, Hungary; balatoni.timea@oncol.hu (T.B.); dr.toth.erika@oncol.hu (E.T.); 3Doctoral College, Semmelweis University, H-1085 Budapest, Hungary; 4Department of Oncodermatology, National Institute of Oncology, H-1122 Budapest, Hungary; 5Center of Radiotherapy, National Institute of Oncology, H-1122 Budapest, Hungary; frohlich.georgina@oncol.hu; 6Department of Biophysics, Eötvös Loránd University, H-1117 Budapest, Hungary; 7Department of Diagnostic Radiology, National Institute of Oncology, H-1122 Budapest, Hungary; bocs.katalin@oncol.hu; 8Department of Surgical and Molecular Pathology, National Institute of Oncology, H-1122 Budapest, Hungary; 9Department of Pathology and Experimental Cancer Research, Semmelweis University, H-1085 Budapest, Hungary; mohos.anita@gmail.com; 10Department of Dermatology, Venereology and Dermatooncology, Semmelweis University, H-1085 Budapest, Hungary; 11Faculty of Medicine, Semmelweis University, H-1085 Budapest, Hungary; 12Faculty of Science, Eötvös Loránd University, H-1117 Budapest, Hungary; 13Department of Surgery, Massachusetts General Hospital, Harvard Medical School, Boston, MA 02115, USA

**Keywords:** melanoma, immunotherapy, PD-1 inhibitors, biomarker, tumor-infiltrating immune cells, HLA class I and II expression

## Abstract

PD-1 inhibitors are known to be effective in melanoma; however, a considerable proportion of patients fail to respond to therapy, necessitating the identification of predictive markers. We examined the predictive value of tumor cell HLA class I and II expression and immune cell infiltration in melanoma patients treated with PD-1 inhibitors. Pretreatment surgical samples from 40 stage IV melanoma patients were studied immunohistochemically for melanoma cell expression of HLA class I molecules (using four antibody clones with different specificities), HLA-II, and immune cell infiltration (using a panel of 10 markers). Among the responders, the ratio of patients showing melanoma cell HLA-II expression was higher compared to non-responders (*p* = 0.0158), and similar results were obtained in the case of two anti-HLA-I antibodies. A combined score of HLA-I/II expression also predicted treatment response (*p* = 0.0019). Intratumoral infiltration was stronger in the responders for most immune cell types. Progression-free survival showed an association with HLA-II expression, the combined HLA score, and the density of immune cells expressing CD134 and PD-1, while overall survival was significantly associated only with HLA class II expression. Our findings corroborate previous results indicating the importance of immune cell infiltration and tumor cell HLA-II expression in the efficacy of PD-1 inhibitor treatment in a “real world” patient cohort and suggest the potential predictive role of HLA class I expression.

## 1. Introduction

Immune checkpoint inhibitors (ICIs), especially antibodies targeting the PD-1/PD-L1 pathway, have transformed cancer treatment, with documented favorable effects in a variety of tumor types, even in advanced stages. However, a considerable proportion of patients (the size of which varies depending on the tumor type) fail to respond to therapy or develop resistance after the initial response. The factors causing either primary or acquired resistance are only partly elucidated, and it is of primary importance to search for biomarkers that could predict the likelihood of therapeutic effects [1,2,3].

Many potential biomarkers have been proposed that could predict the efficacy of ICI therapies. They include, among others, PD-L1 expression in the case of PD-1/PD-L1 targeted agents, tumor mutational burden (TMB), neoantigen load, microsatellite instability, antigen presentation deficiencies, tumor infiltration by immune cells, and immune-related gene expression in tumors [1,2,3,4,5,6,7]. However, few of the above biomarkers have been validated, and most of them were investigated only in one or a few studies. Although malignant melanoma is one of the tumor types with the best response rates to ICIs, more than half of stage IV patients receiving anti-PD-1 therapy still experience treatment resistance [8,9,10]. There is no approved predictive biomarker for PD-1 blocking therapy in metastatic melanoma patients. PD-L1 expression of tumor cells and/or tumor-infiltrating immune cells, which is a prerequisite of anti-PD-1/PD-L1 treatment for several cancer types, is not a clinically validated biomarker in melanoma; it was found to be associated with clinical response or post-therapy survival in most but not all studies [4,11,12,13,14]. Among other components of the tumor immune microenvironment (TIME), infiltration by CD8^+^ T lymphocytes was consistently found to be predictive of response while results concerning CD4^+^ T cells are controversial [4,13,15,16,17,18]. Other immune cell subset markers (e.g., CD16, CD45RO) and immune checkpoints such as PD-1 and CTLA-4 have also been found to be associated with the treatment outcomes in some studies [4,13,18,19].

Since antigen presentation by MHC class I molecules is the prerequisite of tumor cell recognition by cytotoxic T lymphocytes, considered to be major players in antitumor immune responses, it could be logically expected that MHC-I expression by tumor cells would be required for the efficacy of T cell-based immunotherapies, including ICIs. Human cancer cells frequently downregulate the components of HLA class I antigen processing machinery (APM), which was found to be associated with poor prognosis in several tumor types [20]. The mutation or loss of heterozygosity of β2-microglobulin (β2M) was identified as a mechanism of resistance to immunotherapies, including PD-1 inhibitors [7,21]. However, decreased HLA class I expression is most frequently caused by epigenetic or transcriptional mechanisms [20,22]. The potential association of HLA class I protein expression loss with ICI therapy resistance has been explored only to a limited extent [23,24,25]. Surprisingly, the studies demonstrated the predictive role of tumor cell HLA class I expression in melanoma patients treated with the CTLA-4 inhibitor ipilimumab [24,25] but not in those receiving PD-1 blocking agents [23,24]. Conversely, HLA class II expression on tumor cells proved to be predictive of anti-PD-1 efficacy but not anti-CTLA-4 effects [23,24,25].

In our previous studies, we proved the role of HLA class I expression and infiltration by several immune cell types (e.g., CD4^+^ and CD8^+^ T lymphocytes, regulatory T cells, B lymphocytes, NK cells, CD68^+^ macrophages, PD-1^+^ and CD134^+^ lymphocytes) in predicting responses to treatment and/or the survival of melanoma patients receiving ipilimumab therapy [25,26]. The aim of the present study was to explore tumor cell HLA class I and class II expression and tumor-infiltrating immune cell types as potential predictive markers in melanoma patients treated with PD-1 inhibitors.

## 2. Materials and Methods

### 2.1. Patients and Tumor Samples

Archived paraffin blocks of pretreatment surgical samples from 40 stage IV melanoma patients receiving anti-PD-1 therapy (nivolumab 240 mg every two weeks or pembrolizumab 200 mg every 3 weeks, start of treatment in 2015–2022) at the National Institute of Oncology were selected. Only patients receiving at least 3 cycles of anti-PD-1 therapy were included. Tumor resection samples of lymph node (*n* = 70) and skin/subcutaneous metastases (*n* = 42), obtained within 4 years of starting anti-PD-1 therapy (median 10 months, range 1–48), were selected; the study cohort consisted of 40 patients (1–9 examined lesions per patient). The primary site was cutaneous in 38 cases and unknown in 2 cases. Most patients (*n* = 28) received the anti-PD-1 treatment as first-line therapy. Response assessment was based on immune-related response criteria (irRC) [27]; patients with complete or partial remission as best overall response were considered responders in the evaluation. Patients’ follow-up was performed by CT every 3 months and if it was necessary, MRI or PET/CT was also performed. Follow-up time of surviving patients was median 62.5 months (22–91). Progression-free survival (PFS) and overall survival (OS) were defined as the time from commencing anti-PD-1 treatment to disease progression or death or last follow-up, and death or last follow-up, respectively. Patient characteristics are shown in Table 1.

### 2.2. Immunohistochemical Staining

Immunohistochemical staining of tissue sections of formalin-fixed, paraffin-embedded tumor samples was performed utilizing standard methodology as described earlier [25,26]. Briefly, after deparaffination, sections were treated with 3% H_2_O_2_ in methanol to block endogenous peroxidases, followed by antigen retrieval by heating at 98 °C for 40 min in citrate buffer (pH 6.0). After incubation with protein blocking solution (Protein Block, Serum-Free, Dako, Glostrup, Denmark) for 10 min at room temperature, primary antibodies were applied overnight at 4 °C. For the detection of immune cell markers, the following monoclonal antibodies were used: CD8 (C8/144B; Dako), CD45R0 (UCHL1; Dako), CD20y (L26; Dako), FOXP3 (236/E7; eBioscience, San Diego, CA, USA), NKp46/NCR1 (195314; R&D Systems, Minneapolis, MN, USA), CD103 (EPR4166(2); Abcam, Cambridge, UK), CD134 (Ber-ACT35; BioLegend, San Diego, CA, USA), CD137 (BBK-2; Santa Cruz Biotechnology, Dallas, TX, USA), PD-1 (NAT-105; Bio SB, Santa Barbara, CA, USA), PD-L1 (73-10; Abcam). For detecting MHC class I molecules, HCA2 (Origene, Rockville, MD, USA), recognizing β2M-free HLA-A (excluding -A24), -B7301, and -G heavy chains; HC10 (Origene), recognizing β2M-free HLA-A3, -A10, -A28, -A29, -A30, -A31, -A32, -A33, HLA-B (excluding -B5702, -B5804, and -B73) and HLA-C heavy chains; the β2-microglobulin (β2M) specific monoclonal antibody NAMB-1; and the anti-pan HLA class I EMR8-5 (Abcam) were utilized; HLA-DR,DQ,DP was detected using the monoclonal antibody LGII-612.14. Staining was detected using the EnVision+ System HRP Labeled Polymer Anti-mouse reagent (Dako) and the MACH2 Rabbit HRP-Polymer (Biocare Medical, Pacheco, CA, USA) for mouse and rabbit primary antibodies, respectively, followed by staining with 3-amino-9-ethylcarbazole (AEC; Vector Laboratories, Inc., Burlingame, CA, USA) and hematoxylin counterstaining.

### 2.3. Evaluation of the Immune Reactions

Evaluation of immunohistochemical staining was performed independently by two researchers who were blinded to the clinical information, as described earlier [25,26], using light microscope equipped with an eyepiece graticule, and the mean value of their separate counts was used for the analysis. For immune cell evaluation, labeled cells within the metastases were counted in at least 10 randomly chosen fields per section, using the graticule of 10 × 10 squares designating an area of 0.0625 mm^2^ at 400× magnification. PD-L1 expression by immune cells was evaluated similarly to the other immune cell markers, while its expression by tumor cells (tumor proportion score, TPS) was also registered. HLA class I staining was scored as 0, 1, and 2 when the percentage of stained melanoma cells was <25%, 25–75%, and >75%, respectively, based on the criteria established by the 12th International Histocompatibility Workshop (1996). Expression on normal cells in the samples (e.g., immune cells, cells of the vasculature) served as positive control. In the case of HLA-DR,DQ,DP, the percentage of the area displaying positive tumor cell staining was determined in the metastases. We also calculated a combined score of HLA class I and class II expression (HLA I/II score) based on the number of anti-HLA-I antibodies showing higher positivity than the cutoff level, combined with HLA-II positivity in a given sample (score of 1 in the case of high labeling with at least 3 anti-HLA-I antibodies, and/or HLA-II expression ≥3%; score of 0 when neither of the above criteria are met). For patients with more than one metastasis available, the average scores were calculated for each marker. Cutoff levels were set up based on the median of the given variable in the whole patient cohort, with modifications for better discriminating power in some cases, and the proportion of patients with a mean cell density/HLA expression score higher than the cutoff level was also determined for all markers.

### 2.4. Statistical Analysis

For the statistical analysis of differences in cell densities and HLA expression levels between different patient groups, we applied the Mann–Whitney U test, while Fisher’s exact test was used to compare the proportions of cases in different groups. Correlations were analyzed by Pearson’s nonparametric correlation method. Survival analysis was carried out using the Kaplan–Meier method and log-rank test. Differences were considered significant in the case of *p* values ≤ 0.05. Statistics were calculated using the Statistica software version 12.5 (StatSoft, Tulsa, OK, USA).

## 3. Results

Of the 40 stage IV melanoma patients involved in this study, 28 exhibited a complete (*n* = 14) or partial response (*n* = 14) (Table 1). Among clinical parameters, the ECOG performance status and pretreatment LDH level showed an association with the response. Pretreatment surgical samples of lymph node and cutaneous/subcutaneous melanoma metastases were immunohistochemically analyzed for HLA expression (Figure 1a), evaluating it using scores between 0 and 2 for HLA class I and determining the percentage of positive tumor area in the case of HLA class II. Considering all metastases, tumor cell HLA-I positivity levels determined by the four different antibodies (HCA2, HC10, NAMB-1, EMR8-5) showed a strong positive correlation, with the lowest correlation coefficient (0.6812) in the case of HCA2/HC10 and highest (0.8236) in the case of NAMB-1/EMR8-5 (all *p* values < 0.0001), while HLA-II staining showed a weaker correlation with HLA-I positivity (correlation coefficients 0.2182–0.2944). HLA class I expression scores were the highest in the case of the EMR8-5 pan-HLA-I antibody (mean ± SD, 1.4 ± 0.7) and the lowest in the case of HC10 (1.1 ± 0.8). Melanoma cells in metastases of 20 patients (50%) did not express HLA class II, while mean tumor cell staining higher than 10% was observed in 11 patients (27.5%). The tumor samples of eight patients showed 3–4% HLA-II staining, mainly in tumor cells near the inflammatory cells at the margin of metastases, consistent with locally induced HLA expression.

Among the responders, the proportion of patients showing HLA-II expression in ≥3% of melanoma cells was higher compared to non-responders (17/28 vs. 2/12, *p* = 0.0158). Similarly, a significant difference was found in the case of two anti-HLA-I antibodies (HC10 and EMR8-5), with fewer cases showing decreased expression in responders (Table 2). A combined score of HLA-I and -II expression was introduced (being positive in the case of high HLA class I and/or HLA class II expression), which proved very effective in predicting treatment response (*p* = 0.0019) (Table 2).

We also determined the intratumoral density of immune cell subsets in the melanoma metastases using 10 different markers (CD8, CD45RO, CD20, FOXP3, NKp46, CD103, CD134, CD137, PD-1, PD-L1) (Figure 1b). Among the immune cell types, CD45RO^+^ T lymphocytes were present in the highest number, followed by CD8^+^ T lymphocytes, PD-L1^+^, CD103^+^, and PD-1^+^ cells, while NKp46^+^ NK cells and cells expressing the CD134 and CD137 activation markers were the least numerous (Table 3). Evaluating the infiltration of the labeled cells in relation to immunotherapy response revealed that most of the immune cell subsets were present at higher amounts in the metastases of responding patients compared to non-responders (Table 3, Figure 2). Analyzing the proportion of patients with high mean intratumoral cell density showed higher prevalence in responders in the case of eight of the ten immune cell markers (Table 3). The most significant differences were observed in the case of PD-1 (23/28 vs. 3/12, *p* = 0.0010) and PD-L1 (19/28 vs. 1/12, *p* = 0.0012). Interestingly, PD-L1 expression (≥1%) by tumor cells did not predict treatment response significantly (9/28 of responders, 1/12 of non-responders; *p* = 0.2307), in contrast to its expression by immune cells.

Appendix A presents individual patients’ data and a heatmap showing all markers (HLA expression and immune cell density), demonstrating an overall higher HLA expression and immune cell prevalence in responders. A coordinated high expression of at least six of the ten immune cell markers was found in 18 of the 28 responders (64%) and only 1 of the 12 non-responders (8%) (*p* = 0.0015).

Kaplan–Meier survival analysis demonstrated a near-significant (*p* = 0.0530) association of PFS with HLA class I expression detected by the HC10 antibody and a highly significant association (*p* = 0.0025) with HLA class II expression (Figure 3a,b). The combined score of HLA class I and class II expression was also significantly linked to PFS (*p* = 0.0166; Figure 3c). Among immune cell markers, a significant survival correlation was demonstrated in the case of CD134 and PD-1, both showing longer PFS in cases with high immune cell density (*p* = 0.0318 and *p* = 0.0230, respectively; Figure 3d,e). Overall survival was significantly associated only with HLA class II expression (*p* = 0.0126; Figure 3f).

## 4. Discussion

Although ICI therapy has revolutionized the treatment of many cancer types, the majority of patients fail to achieve durable responses, and continued efforts to unravel the mechanisms of immunotherapy resistance are necessary to allow the optimization of therapeutic strategies for patients.

In our study, we focused on investigating local immunological parameters, i.e., tumor cell HLA expression and immune cell infiltration, as potential biomarkers of the efficacy of anti-PD-1 treatment in metastatic melanoma patients. As antigen presentation by MHC-I molecules on target cells is essential for recognition and killing by CD8^+^ cytotoxic T lymphocytes, it is a logical assumption that MHC-I expression by tumor cells would be needed for the efficiency of T cell-based immunotherapies such as ICIs. Surprisingly, however, the few earlier studies aiming at demonstrating the predictive role of tumor cell HLA class I protein expression in patients treated with PD-1 inhibitors could not prove an association with therapy outcomes in melanoma [23,24], although it was found to be predictive in the case of CTLA-4 blocking with ipilimumab [24,25]. On the other hand, HLA-II expression by tumor cells predicted anti-PD-1/PD-L1 therapy efficacy in melanoma and breast cancer, while no such association was found with ipilimumab efficacy in melanoma patients [23,24,25,28].

In our present study on stage IV melanoma patients, a higher level of tumor cell HLA class I expression was detected by two of the four antibodies used: HC10, recognizing mainly HLA-B and -C antigens and the pan-HLA class I EMR8-5; this result is similar to our findings in patients receiving ipilimumab therapy [25]. The discrepancy regarding the predictive role of melanoma cell HLA class I expression between our results and those obtained in previous studies [23,24] may partly be due to methodological differences, for example, using biopsies vs. whole sections; moreover, in one of the previous studies, only HLA-A expression was tested while HLA-B and -C were not [23]. On the other hand, in our cohort, HLA class II expression seemed to have a more significant impact on the treatment outcome, which is in line with the results of previous investigations on melanoma patients receiving anti-PD-1 treatment [23,24] and different from the findings of studies on ipilimumab therapy [24,25]. The different mechanisms of action of anti-CTLA-4 and anti-PD-1 agents may explain the observed dissimilarities in the associations of HLA class I vs. class II tumor cell expression with therapy outcomes; however, the exact mechanisms explaining this divergence are unclear at present.

An additional potential explanation for the stronger association of HLA class II expression with treatment response is that it is highly inducible by cytokines such as IFN-γ, which is mainly produced by T lymphocytes. Consequently, HLA-II expression generally shows a strong correlation with the density of T cells [25,29,30], which is often reported as a factor predicting immunotherapy response, which is also supported by our results; thus, it is possible that the association between HLA-II expression and treatment response results, at least in part, indirectly from the above correlations. In the present study, the associations with either clinical response or survival did not appear to depend on the percentage of HLA-II-positive tumor cells (comparing cases with 3–4% vs. >10% positivity; see Appendix A), which may also support the hypothesis of indirect effect. Another alternative mechanism that may be claimed to be in the background of the impact of HLA-II expression in immunotherapy efficiency is the possibility of the activation of helper and cytotoxic CD4^+^ T lymphocytes by tumor antigens presented by these molecules [31]. MHC-II molecules can bind an even greater diversity of peptides than MHC-I, and a study on 5942 tumors demonstrated that mutated peptides that poorly bound to MHC-II were positively selected during tumorigenesis, which had a stronger effect compared to selective pressure against MHC-I-restricted neoantigens [32,33]. In a clinical trial of personalized neoantigen vaccine plus anti-PD-1 therapy, the successful induction and cytotoxic potential of neoantigen-specific CD4^+^ and CD8^+^ T cells was demonstrated [34]. Furthermore, a study analyzing infused TIL products in melanoma patients receiving adoptive cell therapy revealed that neoantigen-specific T cell clones were predominantly CD4^+^ and displayed cytotoxicity [35].

With regard to the seemingly lower influence of pretreatment tumor cell HLA class I expression (compared to HLA-II expression) on the efficacy of PD-1 blocking agents observed in our and others’ studies, it should be mentioned that in our cohort, there were very few patients with completely or almost completely negative tumor cell staining with the four anti-HLA-I antibodies used, and it is possible that a low level of expression is sufficient for T cell recognition is some cases. Furthermore, the ICI treatment was reported to result in immune activation in the TME, for example, enhanced CD8^+^ T cell infiltration [4,19,36], which could induce HLA class I expression on tumor cells in response to increased lymphocyte cytokine production. The potential changes in TME could not be taken into account in our analysis, which evaluated only pretreatment samples. Nevertheless, we could detect an additional value of determining HLA class I expression in the pretreatment tumor samples since the combined score of HLA-I and HLA-II expression proved a more robust predictor of clinical response than HLA-II expression alone. The same results could be obtained when using a simplified score consisting of the combination of HLA-II expression with HLA-I expression based on labeling by the pan-HLA class I antibody EMR8-5 (instead of on labeling by all four antibodies used), which would provide a biomarker that could be used more easily in routine clinical practice.

Besides tumor cell MHC expression, we also investigated the predictive potential of a panel of immune cell markers and found a significant association with response to anti-PD-1 therapy in the case of eight of the ten immune cell types studied: CD8^+^ and CD45RO^+^ T lymphocytes, regulatory T cells (FOXP3^+^), CD20^+^ B lymphocytes, and cells expressing activation markers/immune checkpoints CD134, PD-1, PD-L1, and CD103, a marker of tissue-resident T cells. Correlations of the prevalence of immune cell subsets with disease outcomes in metastatic melanoma patients after anti-PD-1 therapy have been consistently reported for CD8^+^ T cells [4,13,15,16,17,18] and found in some studies for other immune cell markers, including CD45RO and FOXP3 [13], PD-1, PD-L1 [4,13], and CD103 [37].

To the best of our knowledge, our study is the first that identified the density of CD134^+^ cells as a factor predicting the response to treatment with anti-PD-1 agents in metastatic melanoma patients and also showed an association with PFS. CD134 (OX40) is an important costimulatory immune checkpoint molecule, also a T cell activation marker, expressed mainly by CD4^+^ T cells, also pointing to the potential importance of CD4^+^ T cell activation upon antigen presentation by HLA-II. It shows transient expression after antigen stimulation, which may be one of the reasons for the relatively low number of CD134^+^ cells in tumors; furthermore, they generally show a preferential location in peritumoral/stromal areas while, in our study, intratumoral density was evaluated. The prevalence of CD134^+^ cells was proven to be a prognostic factor in primary melanoma [38] and other cancers [39,40] and a biomarker that could predict the response to ipilimumab therapy [26] in previous studies.

The results regarding CD20^+^ B lymphocytes are controversial; no correlation with response was found in one study [17], although an association with treatment response was demonstrated in a neoadjuvant trial [41]. The role of B cells in antitumor immune responses is also not clear, with controversial findings regarding their pro- vs. antitumor effects. This discrepancy may be related to the diversity of their functional activities; they can promote tumor growth via multiple mechanisms, e.g., driving chronic inflammation, or immune suppression by regulatory B cells, but may also function as effective antigen-presenting cells promoting antitumor T cell responses [42,43]. Costimulation via the OX40L/OX40 pathway has been described as a mechanism of the B lymphocyte-mediated expansion of CD4^+^ T cells in a murine model [44]. In this regard, our earlier findings of the colocalization of B cells and OX40^+^ T lymphocytes in primary melanoma may be of interest, suggesting a possible role of B cells in antigen presentation and costimulation [45].

The densities of most of the studied immune cell types strongly correlated with each other and they frequently showed coordinate presence, which occurred more frequently in the responders. Besides the factors involved in this study, many other elements of the TME may affect the efficacy of immunotherapy, such as macrophages, myeloid-derived suppressor cells, cytokines, and chemokines, among others. According to recent analyses, the application of multiplex immunohistochemistry (IHC)/immunofluorescence techniques yields more accurate predictive markers than studying single factors [18,46]. Multiplex imaging techniques allowing the evaluation of the spatial organization of the tumor microenvironment (TME) are increasingly applied in the search for factors predicting the efficacy of immunotherapy [47,48].

Among the immune cell markers, the most significant association with treatment response was observed in the case of PD-1 and PD-L1; the former showed a correlation with PFS as well. Interestingly, while the intratumoral density of PD-L1-positive immune cells was demonstrated to predict clinical responses to PD-1 blocking therapy in our patient cohort, its expression by melanoma cells did not prove predictive. Evaluating PD-L1 expression by using the CPS (combined positive score) and TPS (tumor proportion score) supported these findings. Previous studies exploring the predictive role of PD-L1 in melanoma patients treated with anti-PD-1 immunotherapy mainly measured its expression by tumor cells, or by tumor and immune cells without distinction [4,11,12,13,14], and did not evaluate the impact of immune cell expression separately. In a study applying digital spatial profiling, PD-L1 expression in the CD68^+^ (macrophage) compartment but not in the tumor compartment proved predictive of immunotherapy effects [49]. In the study by Herbst et al. [50], PD-L1 expression by tumor-infiltrating immune cells, but not by tumor cells showed an association with the response to atezolizumab (anti-PD-L1) treatment in a mixed cohort of different cancer and in the subgroup of NSCLC. Furthermore, according to several studies in mouse models, PD-L1 expression on host myeloid cells, and not on tumor cells, proved essential for PD-L1 pathway-mediated tumor regression [51,52].

The analysis of progression-free survival demonstrated a tendency of better PFS in cases with high HLA class I expression detected by the HC10 antibody. However, significantly longer PFS was demonstrated in cases with high tumor cell expression of HLA class II, high HLA I/II scores, and a high density of PD-1^+^ and CD134^+^ cells. Overall survival, on the other hand, showed a significant association only with HLA-II expression, which was also the factor most significantly influencing PFS. The less prominent associations of the examined factors with OS compared to the clinical response and PFS may be due to the fact that after progression on anti-PD-1 therapy, other treatment modalities were applied in most patients, so their overall survival was not only influenced by the effectivity of PD-1 blocking therapy.

While our study identified several potential biomarkers of responses to PD-1 blocking therapy, we recognize its inherent limitations, mainly caused by its retrospective nature. Moreover, the number of cases included in the analysis was constrained by the availability of sufficient surgical samples and the selection criteria we used in an attempt to decrease patient and sample variability (e.g., including only stage IV melanoma patients, whole sections of surgical samples, and only lymph node and skin/s.c. metastases). On the other hand, we believe that the applied selection criteria enhance the reliability of our results compared to studies performed on samples of unspecified or very heterogeneous locations and studies with a wide range of time intervals between sample acquisition and the implementation of immunotherapy. Also, the use of whole sections vs. small biopsies (or tissue microarrays) could reduce the impact of intratumoral heterogeneity, which is important in the case of immune markers that are often highly heterogeneously distributed within the tissue.

A further limitation of our study is that it included only pretreatment tumor samples, intending to identify potential predictive markers. However, the tumor microenvironment is dynamically changing during progression and therapies, and in order to explore the reasons for acquired resistance to immunotherapy, longitudinal analyses would be beneficial. In our previous investigation comparing HLA class I expression and immune cell infiltration in pre- and post-treatment tumor samples of melanoma patients receiving ipilimumab therapy, we found HLA-I downregulation in the progressing metastases of non-responding patients [53]. Our future plans include using this approach in a longitudinal study of patients in the present cohort treated with PD-1 inhibitors.

## 5. Conclusions

Summarizing our findings, they corroborate previous results, indicating the importance of tumor infiltration by CD8^+^ T lymphocytes, PD-1^+^ and PD-L1^+^ cells, as well as immune cell subsets that are less frequently studied, such as CD45RO^+^ memory T cells, FOXP3^+^ regulatory T cells, CD103^+^ tissue-resident T cells, CD20^+^ B lymphocytes, in the efficacy of PD-1 blocking therapy. This study also proposes the density of T cells expressing CD134 as a potential new biomarker in a “real world” patient cohort. A coordinated high expression of the immune cell markers was also strongly associated with the treatment response. Furthermore, our results support earlier reports on the predictive value of tumor cell HLA class II expression and suggest the potential additive value of the HLA-I expression level.

## Figures and Tables

**Figure 1 biomolecules-14-01609-f001:**
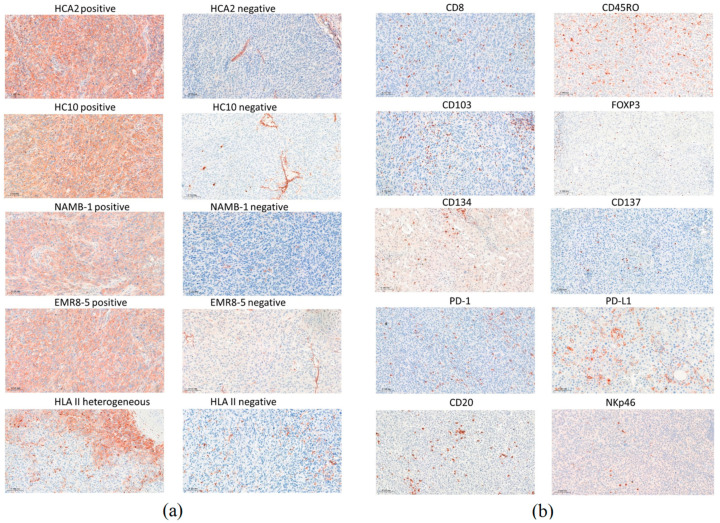
Immunohistochemical staining of melanoma metastases with HLA class I and class II-specific antibodies (**a**) and labeling of immune cell markers (**b**) (3-amino-ethylcarbazole, red). Scale bars: 100 μm.

**Figure 2 biomolecules-14-01609-f002:**
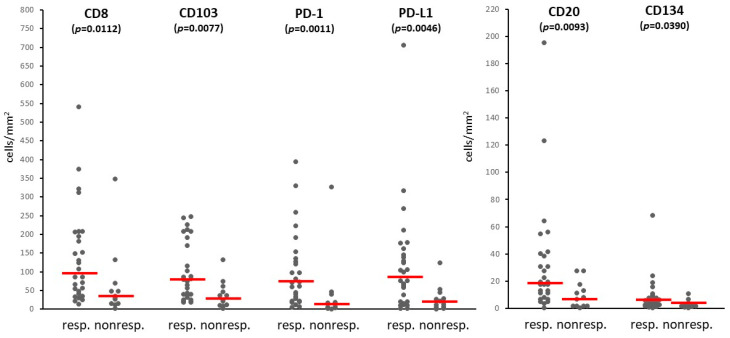
Immune cell infiltration in metastases of responder and non-responder patients treated with PD-1 inhibitors (dots: mean density values in samples from individual patients; horizontal line: median).

**Figure 3 biomolecules-14-01609-f003:**
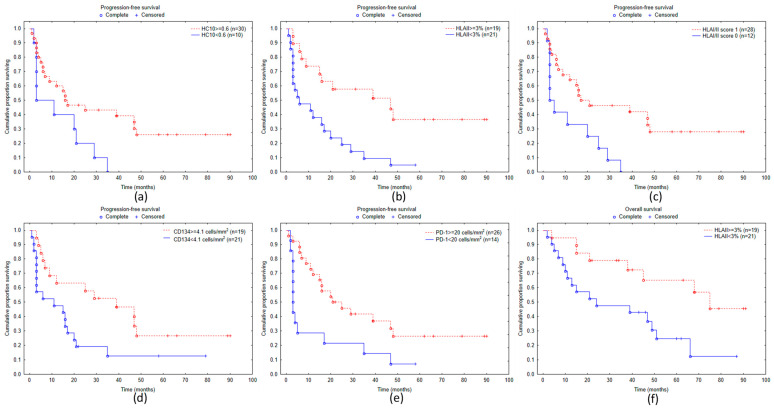
Kaplan–Meier curves of progression-free survival for melanoma patients subdivided according to staining with HLA class I-specific antibody HC10 (**a**), tumor cell HLA-II expression (**b**), HLA I/II score (**c**), density of CD134^+^ (**d**), PD-1^+^ cells (**e**), overall survival according to tumor cell HLA class II expression (**f**).

**Table 1 biomolecules-14-01609-t001:** Patient and sample characteristics.

	Responder (CR, PR) *n* = 28	Non-Responder (SD, PD) *n* = 12	*p* Value
Age, years: median (range)	62 (27–87)	53 (34–77)	NS ^a^
Gender			
female	11	2	
male	17	10	NS ^b^
ECOG performance status			
0	27	8	
1	1	4	0.0223 ^b^
BRAF mutation status			
wild type	20	10	
mutant	8	2	NS ^b^
LDH level			
normal	23	4	
>ULN	5	8	0.0075 ^b^
Anti-PD-1 drug			
nivolumab	9	8	
pembrolizumab	19	4	NS ^b^
Line of therapy			
1	20	8	
2–5	8	4	NS ^b^
PFS, months: median (range)	27 (4–90+)	3 (1–11)	0.0000 ^a^
OS, months: median (range)	46 (13–91+)	9 (4–62+)	0.0002 ^a^

^a^ Mann–Whitney test, ^b^ Fisher’s exact test. CR: complete response, PR: partial response, SD: stable disease, PD: progressive disease, ECOG: Eastern Cooperative Oncology Group, BRAF: B-Raf proto-oncogene, LDH: lactate dehydrogenase, PFS: progression-free survival, OS: overall survival, ULN: upper limit of normal, NS: not significant.

**Table 2 biomolecules-14-01609-t002:** Relationship of treatment response with the proportion of patients with high tumor cell HLA expression in metastases.

	Responder (CR, PR) *n* = 28	Non-Responder (SD, PD) *n* = 12	*p* Value ^a^
HLA class I; Ab clone (cutoff score)			
HCA2 ^b^ (≥1.3)	16/27 (59%)	5/12 (42%)	0.4877
HC10 (≥0.6)	24/28 (86%)	6/12 (50%)	0.0411
NAMB-1 (≥1.7)	14/28 (50%)	3/12 (25%)	0.1788
EMR8-5 (≥1.4)	20/28 (71%)	4/12 (33%)	0.0367
HLA class II (cutoff: ≥3%)	17/28 (61%)	2/12 (17%)	0.0158
HLA I/II score ^c^ (cutoff: ≥3)	24/28 (86%)	4/12 (33%)	0.0019

^a^ Fisher’s exact test. ^b^ Staining with HCA2 could not be evaluated in one patient because of negativity of normal cells. ^c^ Combined score of high expression with the anti-HLA-I and/or the anti-HLA-II antibodies. CR: complete response, PR: partial response, SD: stable disease, PD: progressive disease.

**Table 3 biomolecules-14-01609-t003:** Relationship of treatment response with the density of immune cells infiltrating metastases.

Density of Labeled Cells (*n*/mm^2^), Median (Range)			
Immune Cell Marker	Responder (CR, PR) *n* = 28	Non-Responder (SD, PD) *n* = 12	*p* Value ^a^
CD8	96.0 (12.8–540.8)	31.2 (1.6–347.2)	0.0112
CD45RO	416.8 (64.0–769.6)	194.4 (83.2–774.4)	0.1918
CD20	18.4 (0.0–195.2)	7.2 (0.0–27.2)	0.0093
NKp46	3.0 (0.0–16.3)	1.3 (0.0–10.6)	0.3132
FOXP3	24.0 (4.8–100.8)	16.0 (6.4–108.8)	0.1084
CD134	4.9 (0.0–68.0)	2.5 (0.0–10.7)	0.0390
CD137	7.4 (1.1–37.3)	3.8 (0.2–17.0)	0.2992
CD103	76.0 (17.6–246.4)	29.6 (1.6–131.2)	0.0077
PD-1	71.2 (4.8–393.6)	13.6 (0.0–326.4)	0.0011
PD-L1	86.4 (1.6–705.6)	16.8 (0.0–123.2)	0.0046
**Proportion of Cases with High Cell Density (%)**			
**Immune Cell Marker (Cutoff)**	**Responder (CR, PR)** ***n* = 28**	**Non-Responder (SD, PD)** ***n* = 12**	***p* Value ^b^**
CD8 (≥54 cells/mm^2^)	20/28 (71%)	3/12 (25%)	0.0130
CD45RO (≥264 cells/mm^2^)	18/28 (64%)	3/12 (25%)	0.0378
CD20 (≥28 cells/mm^2^)	10/28 (36%)	0/12 (0%)	0.0188
NKp46 (≥3.4 cells/mm^2^)	13/28 (46%)	3/12 (25%)	0.2969
FOXP3 (≥20 cells/mm^2^)	18/28 (64%)	3/12 (25%)	0.0378
CD134 (≥4.1 cells/mm^2^)	17/28 (61%)	2/12 (17%)	0.0158
CD137 (≥6.7 cells/mm^2^)	14/28 (50%)	3/12 (25%)	0.1788
CD103 (≥36 cells/mm^2^)	21/28 (75%)	4/12 (33%)	0.0297
PD-1 (≥20 cells/mm^2^)	23/28 (82%)	3/12 (25%)	0.0010
PD-L1 (≥54 cells/mm^2^)	19/28 (68%)	1/12 (8%)	0.0012

^a^ Mann–Whitney test, ^b^ Fisher’s exact test. CR: complete response, PR: partial response, SD: stable disease, PD: progressive disease.

## Data Availability

Data supporting the reported results can be found in the Appendix A or provided by the corresponding author upon reasonable request.

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
