# Peer review of "Tumor-Infiltrating Immune Cells and HLA Expression as Potential Biomarkers Predicting Response to PD-1 Inhibitor Therapy in Stage IV Melanoma Patients"

_biomolecules, 2024, doi:10.3390/biom14121609_

Round 1
Reviewer 1 Report
Comments and Suggestions for Authors
This study contributes valuable insights into the immunological underpinnings of PD-1 inhibitor response in stage IV melanoma patients. By expanding the scope of research and addressing the limitations noted, future work could enhance the utility of these biomarkers in clinical decision-making, leading to more personalized and effective treatment strategies for melanoma patients.
- The study mainly looks at biopsy samples taken before treatment begins. By also examining samples during and after treatment, researchers could better understand how the tumor environment changes over time. While I understand it could be difficult to include at this point at least addressing it in discussion would be beneficial. This could help reveal why some patients continue to respond to treatment over a long period, while others may develop resistance.
- The study might consider looking computationally at additional factors that could affect how well immunotherapy works. Instead of just focusing on HLA molecules and common immune cells, the research could also examine other elements like myeloid-derived suppressor cells (MDSCs) or specific types of signaling proteins called cytokines. Including these in the study could provide a more complete picture of what affects the treatment's success.
- The study finds links between certain biomarkers and treatment results, but it doesn't explore why or how these biomarkers affect the outcomes. To understand how immune profiles interact with PD-1 inhibitors, as a bioinformatician I suggest either of the following, you can undertake several research approaches that combine experimental data with computational analysis:
o Molecular Docking and Dynamics Simulations: Utilize molecular docking to predict how PD-1 inhibitors interact at the molecular level with proteins expressed within the immune profiles identified. Molecular dynamics simulations can further elucidate the stability of these interactions over time and under different physiological conditions.
o Pathway Analysis: Deploy bioinformatics tools to analyze signaling pathways that involve PD-1 and its inhibitors. By integrating transcriptomics or proteomics data, you can identify which pathways are activated or repressed in the presence of PD-1 inhibitors and how these changes correlate with the immune profiles.
o Gene Expression Profiling: Use RNA-seq or microarray data to examine changes in gene expression associated with successful PD-1 inhibitor treatment. Bioinformatics software can help in identifying upregulated or downregulated genes in patients responding to treatment compared to non-responders.
Author Response
We express our sincere thanks for the reviewers’ comments which helped to improve the interpretation of our data. According to the suggestions, we made several changes, including modifications in Results and mainly in Discussion. Changes are highlighted with yellow. Point-by-point response to the reviewers’ comments are as follows:
Reviewer 1
- The study mainly looks at biopsy samples taken before treatment begins. By also examining samples during and after treatment, researchers could better understand how the tumor environment changes over time. While I understand it could be difficult to include at this point at least addressing it in discussion would be beneficial. This could help reveal why some patients continue to respond to treatment over a long period, while others may develop resistance.
Response: Thank you for pointing out this important aspect. Our study was retrospective (which is mentioned in Discussion as one of its limitations), performed on archived, surgically removed pretreatment tumor samples, and it did not include biopsies taken at any time during treatment. Longitudinal studies would indeed help in understanding the reasons of acquired resistance (which was not an aim of the present study), and we address this point in the revised manuscript (Discussion, lines 404-412).
- The study might consider looking computationally at additional factors that could affect how well immunotherapy works. Instead of just focusing on HLA molecules and common immune cells, the research could also examine other elements like myeloid-derived suppressor cells (MDSCs) or specific types of signaling proteins called cytokines. Including these in the study could provide a more complete picture of what affects the treatment's success.
Response: The reviewer is right, many other factors may also influence the efficacy of immunotherapy, and studies using multiplex immunofluorescence techniques, for example, could yield more complex picture, as mentioned in Discussion (lines 357-361). However, similarly to many studies, ours was based on conventional immunohistochemistry, and it involved a larger panel of markers compared to the majority of such studies, including immune checkpoints and activation markers beside HLA molecules (detected by 5 different antibodies) and the major immune cell types. Our study could not and did not aim to provide a comprehensive analysis of all possible factors. We expanded the Discussion with referring to the complexity of TME and other factors that might influence the efficacy of immunotherapy (lines 355-357).
- The study finds links between certain biomarkers and treatment results, but it doesn't explore why or how these biomarkers affect the outcomes. To understand how immune profiles interact with PD-1 inhibitors, as a bioinformatician I suggest either of the following, you can undertake several research approaches that combine experimental data with computational analysis:
o Molecular Docking and Dynamics Simulations: Utilize molecular docking to predict how PD-1 inhibitors interact at the molecular level with proteins expressed within the immune profiles identified. Molecular dynamics simulations can further elucidate the stability of these interactions over time and under different physiological conditions.
o Pathway Analysis: Deploy bioinformatics tools to analyze signaling pathways that involve PD-1 and its inhibitors. By integrating transcriptomics or proteomics data, you can identify which pathways are activated or repressed in the presence of PD-1 inhibitors and how these changes correlate with the immune profiles.
o Gene Expression Profiling: Use RNA-seq or microarray data to examine changes in gene expression associated with successful PD-1 inhibitor treatment. Bioinformatics software can help in identifying upregulated or downregulated genes in patients responding to treatment compared to non-responders.
Response: Thank you for the suggestions. However, the main mechanism of action of PD-1 inhibitors (monoclonal antibodies against PD-1) is well known, so we feel that it is not necessary to further explore potential other molecular interactions. Anti-PD-1 antibodies bind to PD-1 with high specificity, thus blocking interaction of it with its ligands PD-L1 and PD-L2. Both nivolumab and pembrolizumab are of the IgG4 immunoglobulin subclass, which has low affinity to complement and Fc receptors. Therefore, it is very likely that the immune markers we examined do not directly interact with the PD-1 inhibitors (with the exception of PD-1, of course), and they may influence the efficacy of the therapy because the baseline expression of HLA and the prevalence/activation state of immune cells determine how effective immune reaction can develop in response to ICI therapy. With regard to the 3rd suggested option, our study focused on determining protein expression in order to get a closer picture of the actual status of the tumor immune microenvironment (since post-transcriptional changes may also modify protein expression which is not taken in account in gene expression analyses). Furthermore, we do not have RNA samples of the tumor tissues involved in the study, so we might only perform gene expression profiling analysis on available datasets of ICI-treated melanomas, and such analyzes have been reported in many studies (for example, references 13, 36 in the present manuscript, as well as other reports such as doi: 10.1158/0008-5472.CAN-16-3556, doi: 10.1038/s41591-019-0654-5, doi: 10.1136/jitc-2020-000974, doi: 10.3389/fimmu.2021.685370, doi: 10.1038/s41467-020-15726-7, doi: 10.1038/s41525-021-00169-w, doi: 10.3390/genes11040435 etc.).
Reviewer 2 Report
Comments and Suggestions for Authors
The manuscript by Hegyi et al describe an immunohistochemical analysis of a group of melanoma patients that have been treated with anti-PD-1 therapy. The authors are comparing responders vs nonresponders for MHC expression and immune cell infiltrate as biomarkers for outcome of therapy. The authors conclude that immune cell infiltrates and tumor cell MHC class II expression are indicators of PD-1 checkpoint efficacy. While the conclusions are generally supported by the data, there are a few questions that should be addressed to enhance the message the authors are giving.
1) The authors state “Among the responders the proportion of patients showing HLA-II expression in ≥3% 193 of melanoma cells was higher compared to nonresponders (17/28 vs. 2/12, p=0.0158). While the results are statistically significant, isn’t ≥3% a rather low bar? How did the authors arrive at this cutoff? What does this mean biologically? Does it mean that relatively few MHC class II expressing melanoma cells would lead to the events that promote clinical responses? What if the bar was raised to ≥5%, would the differences still be statistically significant? These issues should be addressed in the discussion.
2) Another area concerning biological relevance is the absolute density/frequency of various markers. For example, CD134 expression correlated with progression free survival. However, what does 4.9/mm2 compared to 2.5/mm2 mean biologically? CD134 is OX40 which is a costimulatory molecule generally found on CD4+ T cells. This fits with the MHC class II data and could be important to the field. These topics should be addressed in the discussion.
3) Given that the therapeutic efficacy of PD-1 checkpoint has been correlated with high mutation load, what do the authors results mean in terms of melanoma cells presenting neoantigens via MHC class II. There are some interesting implications here that have not been explored in the discussion.
4) The correlation with CD20+ cells is interesting. More speculation on the potential role of B cells in the discussion is warranted.
Author Response
Reviewer 2
1) The authors state “Among the responders the proportion of patients showing HLA-II expression in ≥3% of melanoma cells was higher compared to nonresponders (17/28 vs. 2/12, p=0.0158). While the results are statistically significant, isn’t ≥3% a rather low bar? How did the authors arrive at this cutoff? What does this mean biologically? Does it mean that relatively few MHC class II expressing melanoma cells would lead to the events that promote clinical responses? What if the bar was raised to ≥5%, would the differences still be statistically significant? These issues should be addressed in the discussion. – Response: There is no established cutoff level for tumor cell HLA class II expression, with quite inconsistent thresholds applied according to the related literature (e.g., 1% in ref. 24 and 5% in ref. 23; we used both 1% and 10% in our previous study on ipilimumab-treated patients, ref. 25, which didn’t show association with disease outcome in the case of any cutoffs). In the present study we chose 3% because a relatively large number of tumors showed 3-4% positivity (as we mentioned in the Results section of revised manuscript, lines 184-186), which seemed a reasonable cutoff, probably yielding less uncertain results then the evaluation of staining as low as 1% (which occurred in only one patient). Raising the bar to 5% would omit all these cases, and would not yield significant difference with regard to clinical response. Since it seems that the level of expression (3-4% or above 10%) did not make any difference with regard to either response or survival (see in Table S1), we think that the choice of cutoff level was optimal. The issue was also addressed in Discussion (lines 287-290).
2) Another area concerning biological relevance is the absolute density/frequency of various markers. For example, CD134 expression correlated with progression free survival. However, what does 4.9/mm2 compared to 2.5/mm2 mean biologically? CD134 is OX40 which is a costimulatory molecule generally found on CD4+ T cells. This fits with the MHC class II data and could be important to the field. These topics should be addressed in the discussion. – Response: OX40 is generally expressed transiently upon activation, and preferentially in stromal areas, while we evaluated intratumoral density; these factors may contribute to the relatively low density observed in our study. Nevertheless, it appears to function as a biomarker in our study (and as prognostic marker in several tumor types). These issues are discussed in the revised manuscript (lines 332-339).
3) Given that the therapeutic efficacy of PD-1 checkpoint has been correlated with high mutation load, what do the authors results mean in terms of melanoma cells presenting neoantigens via MHC class II. There are some interesting implications here that have not been explored in the discussion. – Response: We included some discussion about this topic in the revised manuscript (lines 293-302).
4) The correlation with CD20+ cells is interesting. More speculation on the potential role of B cells in the discussion is warranted. – Response: We included more discussion on the topic (lines 343-352)
Round 2
Reviewer 1 Report
Comments and Suggestions for Authors
Thank you for addressing all the asked concerns